

# Chest X-ray pneumothorax segmentation using U-Net with EfficientNet and ResNet architectures

Ayat Abedalla[1,*], Malak Abdullah[1,*], Mahmoud Al-Ayyoub[1] and Elhadj Benkhelifa[2]

[1] Computer Science, Jordan University of Science and Technology, Irbid, Jordan
[2] Smart Systems, AI and Cybersecurity Research Centre, Staffordshire University, Stoke on Trent, UK
* These authors contributed equally to this work.

## ABSTRACT

Medical imaging refers to visualization techniques to provide valuable information about the internal structures of the human body for clinical applications, diagnosis, treatment, and scientific research. Segmentation is one of the primary methods for analyzing and processing medical images, which helps doctors diagnose accurately by providing detailed information on the body's required part. However, segmenting medical images faces several challenges, such as requiring trained medical experts and being time-consuming and error-prone. Thus, it appears necessary for an automatic medical image segmentation system. Deep learning algorithms have recently shown outstanding performance for segmentation tasks, especially semantic segmentation networks that provide pixel-level image understanding. By introducing the first fully convolutional network (FCN) for semantic image segmentation, several segmentation networks have been proposed on its basis. One of the state-of-the-art convolutional networks in the medical image field is U-Net. This paper presents a novel end-to-end semantic segmentation model, named Ens4B-UNet, for medical images that ensembles four U-Net architectures with pre-trained backbone networks. Ens4B-UNet utilizes U-Net's success with several significant improvements by adapting powerful and robust convolutional neural networks (CNNs) as backbones for U-Nets encoders and using the nearest-neighbor up-sampling in the decoders. Ens4B-UNet is designed based on the weighted average ensemble of four encoder-decoder segmentation models. The backbone networks of all ensembled models are pre-trained on the ImageNet dataset to exploit the benefit of transfer learning. For improving our models, we apply several techniques for training and predicting, including stochastic weight averaging (SWA), data augmentation, test-time augmentation (TTA), and different types of optimal thresholds. We evaluate and test our models on the 2019 Pneumothorax Challenge dataset, which contains 12,047 training images with 12,954 masks and 3,205 test images. Our proposed segmentation network achieves a 0.8608 mean Dice similarity coefficient (DSC) on the test set, which is among the top one-percent systems in the Kaggle competition.

Corresponding author
Malak Abdullah,
mabdullah@just.edu.jo

## INTRODUCTION

Medical imaging is a useful and essential technique to understand and visualize the human body's internal structure for clinical applications, diagnostic, treatment, and scientific research purposes (*Abdallah & Alqahtani, 2019*). Medical image segmentation has a crucial role in analyzing and processing medical imaging by dividing the image into several meaningful regions based on pixel characteristics, such as intensity, color, and texture (*Khalid, Ibrahim & Haniff, 2011*; *Yang et al., 2002*). There are several methods and techniques for medical image segmentation. The most general practice is manual segmentation (*Fasihi & Mikhael, 2016*) that experts perform. However, this method is time-consuming, requires trained experts, and suffers from the difference in the segmentation results from expert to expert. Deep learning models, especially deep convolutional neural networks (CNNs) (*LeCun et al., 1998*), have attained great success in various computer vision tasks, such as image classification and image segmentation. CNN has effective techniques that enable it to treat image segmentation as a semantic segmentation, which refers to the task of understanding an image at pixel level by assigning a class label to each pixel of the image (*Ciresan et al., 2012*; *Long, Shelhamer & Darrell, 2015*; *Tran, 2016*). In *Long, Shelhamer & Darrell (2015)*, the researchers proposed a fully convolutional network (FCN), a landmark in semantic segmentation that most modern methods rely on. In *Ronneberger, Fischer & Brox (2015)*, the researchers suggested an encoder-decoder network of FCN called U-Net for biomedical image segmentation. The U-Net architecture utilizes the skipped connections to enable precise pixel-level localization. The success of the U-Net network in the field of medical image segmentation has attracted the attention of many researchers recently (*Dong et al., 2017*; *Li et al., 2018*; *Kumar et al., 2018*).

This paper proposes a novel end-to-end semantic segmentation model, named Ens4B-UNet, for medical images that ensembles four U-Net architectures of different pre-trained Backbones. We evaluate and test our model on the chest radiograph dataset from the 2019 SIIM-ACR Pneumothorax Segmentation Challenge. (https://www.kaggle.com/c/siim-acr-pneumothorax-segmentation/overview) Undoubtedly, pneumothorax is a life-threatening condition that occurs when air present in the pleural cavity between the lungs and the chest wall. Our model determines whether the patient has a pneumothorax and determines its location and extent. The Ens4B-UNet relies on chest X-rays as it is the most common diagnostic method for pneumothorax. It utilizes U-Net's success with several significant improvements by adapting powerful and robust deep CNN models as the backbone network for the encoder and using the nearest-neighbor up-sampling in the decoder. The backbone networks used in our proposed model are 50-layer ResNet (*He et al., 2016*), 169-layer DenseNet (*Huang et al., 2017*), 50-layer ResNext (*Xie et al., 2017*) with the Squeeze and Excitation (SE) block (*Hu, Shen & Sun, 2018*), and EfficientNet-B4 (*Tan & Le, 2019*), all of which are pre-trained on the ImageNet dataset to exploit the benefit of transfer learning. Thus, four segmentation networks have been formed: ResNet50-UNet, DenseNet169-UNet, SE-ResNext50-UNet, and EfficientNetB4-UNet; each one is trained separately. We have used data augmentation and stochastic weight

averaging (SWA) (*Izmailov et al., 2018*) procedure for training. The prediction maps are generated by applying the test-time augmentation (TTA) (*Wang et al., 2018*) and ensembling the outputs from different segmentation networks. Finally, the binary mask was obtained by applying different types of optimal thresholds.

The rest of this paper is organized as follows: "Background" presents detailed background information about medical imaging, pneumothorax, and deep learning. "Related Works" presents the related work of medical image segmentation. "Method" describes our method thoroughly, including dataset description, pre-processing steps with data augmentation, the segmentation network architectures, training procedure, and post-processing strategy. "Experiments" describes our experiments. "Results and Discussion" reports the results and discussion of the proposed model. Finally, the conclusion is drawn in "Conclusion and Future Work".

## BACKGROUND

### Medical imaging

Medical imaging is the process of creating images of the human body's internal structures, which can assist the diagnosis, study, monitor, and treatment of patients (*Abdallah & Alqahtani, 2019*). Different technologies of medical imaging include various radiographic techniques such as X-ray radiography, computed tomography (CT), magnetic resonance imaging (MRI), ultrasound, and elastography, which contain a region of interest (ROI) that doctors use to diagnose diseases. X-ray (*Spiegel, 1995*) is the oldest imaging technology (discovered in 1895) that uses ionizing radiation to create black-and-white images of internal structures of the body. X-rays are absorbed in various quantities depending on the density of the body that passes through. Objects with a high density that block the radiation appear in white, while objects with a low density allow the radiation to penetrate through appearing in black. For example, since the lungs contain air, their densities are low, allowing radiation to pass through; therefore, the image seems more black. In contrast, the bones appear white because they are of high density, blocking more amounts of radiation. The current research focuses on X-ray images. CT (*Abdallah & Alqahtani, 2019*) is a vital imaging technology that uses special rotating X-ray devices and computers to produce multiple slices in different directions that are processed to create cross-sectional (tomographic) images of specific areas within the body. CT scan is useful for visualizing blood vessels, soft tissues, brain, knee, and cancers such as lung cancer and liver cancer. MRI imaging (*Hanson, 2009*) is based on sophisticated technology that uses magnetic fields, radio waves, and a computer to create detailed images of organs and tissues within the body. MRI scanners are often used for the brain, liver, spinal cord, blood, and heart. Ultrasound (*Abdallah & Alqahtani, 2019*) imaging technology uses sound waves with high frequency to produce images of the inside of the body. Ultrasound imaging is widely used in examining the liver, gallbladder, spleen, pancreas, heart, and uterus.

### Pneumothorax

Pneumothorax or collapsed lung is the presence of air in the pleural cavity between the lungs and the chest wall (*Noppen & De Keukeleire, 2008*; *Lugo & Carr, 2019*). The lungs are

covered with two types of pleura: the inner pleura (visceral pleura) that sticks to the lung to protect its external surface, and the outer pleura (parietal pleura) attached to the chest wall allows the lungs to float freely. The pleural cavity is the cavity between visceral and parietal pleurae. During breathing, the visceral and parietal pleurae stick together, and the air exits through the exhalation. But with pneumothorax, the air accumulates in the pleural cavity, leading to pressure on the lungs and, consequently, the lungs' collapse. Pneumothorax is one of the common respiratory diseases that can occur spontaneously or by trauma.

In non-emergency cases, pneumothorax can be diagnosed by clinical examination, using a stethoscope, noting that breathing sounds are low or absent on the affected lung, or clicking on the chest and noting the sound is like a drum. On the other hand, the pneumothorax diagnosis can be considered complex or challenging to explore, especially in small pneumothoraces. Therefore, medical imaging is an essential technique and convenient method for most diagnoses. Chest X-rays are usually obtained to check for pneumothorax existence. If the diagnosis with a chest x-ray is inconclusive, then a chest computed tomography (CT) scan or/and Ultrasound imaging can help identify pneumothorax.

The pneumothorax treatment depends on the patient's symptoms, the presence of the underlying lung disease, the size and type of pneumothorax, and the severity of the condition. The treatment varies from early observation and follow-up to needle decompression, chest tube insertion, no surgical repair, or surgery (*Noppen & De Keukeleire, 2008*). If the pneumothorax area is small, it is usually healed without treatment and does not cause serious problems. It requires monitoring the condition, then the excess air will be completely absorbed in several days and the patient can be given oxygen. While if the area affected by pneumothorax is large, needle aspiration or chest tube is used. Furthermore, in traumatic pneumothorax, chest tubes are usually inserted into the air-filled space to remove air and re-expand the lung. However, in the tension pneumothorax, urgent needle decompression is inserted, and the chest tube is immediately placed (*Zarogoulidis et al., 2014*).

## Deep learning

Deep learning, which is the most powerful machine learning technique nowadays, can automatically discover and learn complex features and patterns from data without human intervention. Deep learning models are built using the artificial neural network (ANN) structure, which is inspired by the human brain structure with neuron nodes connected via synapses (*Goodfellow, Bengio & Courville, 2016*). Moreover, deep learning is constructed using multiple layers to extract features from raw input data. Figure 1 shows an illustration for deep learning structure.

Deep learning algorithms and applications have remarkably achieved great success in many tasks and fields, such as computer vision, natural language processing (NLP) (*Wolf et al., 2019*), and speech recognition (*Saon et al., 2017*). The most popular types of deep learning models are recurrent neural networks (RNNs) and convolutional neural networks (CNNs). RNN, including Long Short-Term Memory (LSTM) (*Hochreiter & Schmidhuber,*

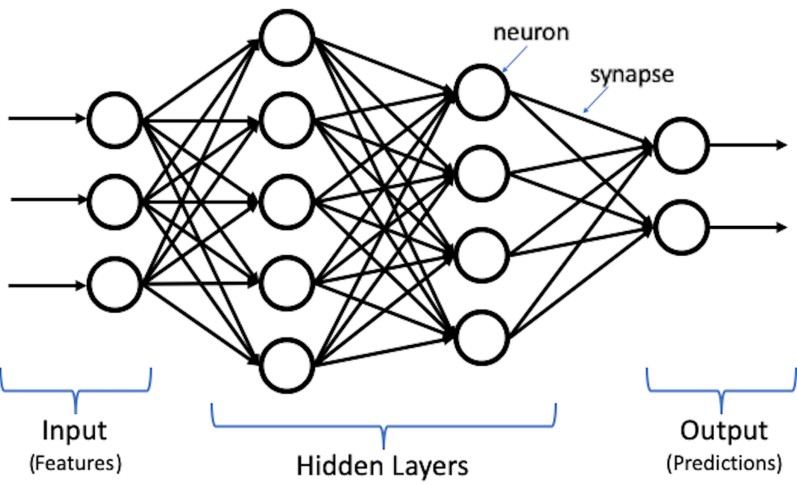

**Figure 1  Illustration of deep learning structure.**

*1997*) and Gated Recurrent Units (GRU) (*Cho et al., 2014*), is defined as a sequence model that handles and processes sequential data using its internal memory and allows information to flow back and forth. It is widely employed in various applications in different domains, such as speech recognition (*Graves & Jaitly, 2014*), image captioning (*Xu et al., 2015*), and machine translation (*Sutskever, Vinyals & Le, 2014*). While CNN is a feed-forward neural network where the information flows in the forward direction to learn its spatial features. It is mostly used for computer vision problems, such as image classification (*He et al., 2016*), object detection (*Ren et al., 2015*), object segmentation (*Long, Shelhamer & Darrell, 2015*; *Ronneberger, Fischer & Brox, 2015*; *Badrinarayanan, Kendall & Cipolla, 2017*), and image style transfer (*Gatys, Ecker & Bethge, 2016*).

CNN's strength lies in several factors, namely, parameter sharing and the sparsity of connections between layers where there is no need to connect every neuron to every other to carry information throughout the network. It mainly consists of convolutional layers, pooling layers, activation layers, and fully-connected layers. In the convolutional layers, the input is convolved using filters. Each filter is a matrix of pixels applied across data through a sliding window. The convolution operation involves taking the element-wise product of filters in the image and then summing those values for every sliding action to output a feature map. The activation function is then applied to help the network use relevant information and terminates the irrelevant information. After that, the output feature map is fed to the pooling layer, which reduces the representation size, thus reducing the computation in the network. Finally, after a series of convolutional and max-pooling layers, the output behaves as high-level input features. These features are now fed into fully connected layers to learn the relationship between the learned features and operate the classification. It is worth mentioning that in the lower layers of a CNN, the network can detect high-level patterns from the image, such as edges and curves. As the network performs more convolutions, it begins to define complex features and specific objects. Figure 2 shows an illustration of the CNN structure.

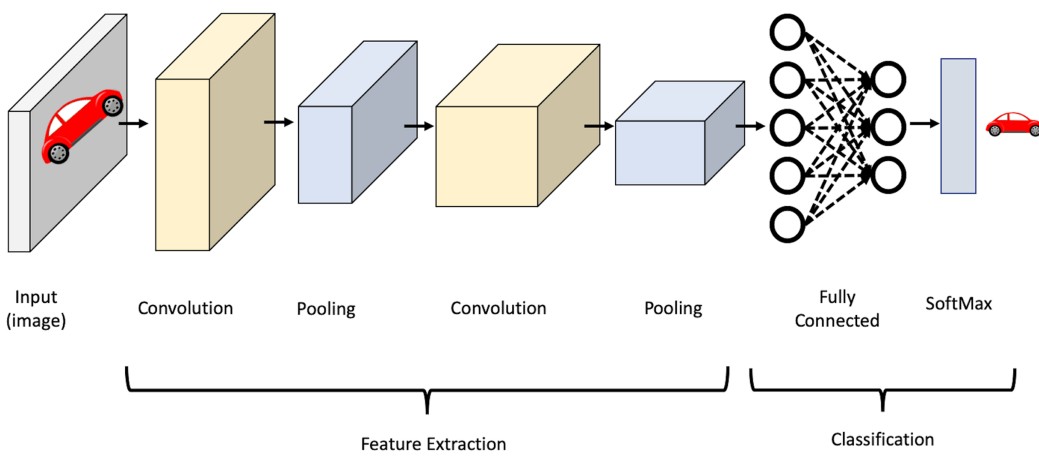

**Figure 2 Illustration of CNN structure.**

It should be noted that CNN became popular in the computer vision (CV) domain in late 2012. It is when a CNN-based network named AlexNet (*Krizhevsky, Sutskever & Hinton, 2012*) won the 2012 ImageNet Large Scale Visual Recognition (ILSVRC) (*Deng et al., 2009*). The researchers of CV community, thereafter, directed their attention to CNN and expanded the experiments to develop various image processing techniques, including VGGNet (*Simonyan & Zisserman, 2014*), GoogleNet (*Szegedy et al., 2015*), ResNet, DenseNet, ResNeXt, EfficientNet, U-Net, SegNet (*Badrinarayanan, Kendall & Cipolla, 2017*), etc.

In 2015, a research work (*Ronneberger, Fischer & Brox, 2015*) developed an extended network of FCN called U-Net for biomedical image segmentation. Being based on an FCN network indicates that the network contains only convolutional layers, which reduce the number of parameters and accept an image of any size. The U-Net is trained in an end-to-end manner and proved to be a highly effective technique in different tasks where the output size is similar to the input size. The network architecture consists of two paths, a contracting path (encoder) to extract features and capture the context in the image and an expansive track (decoder) to enable precise localization. That makes the network layout looks like a U shape. The contracting path consists of repeated application of two 3 × 3 convolutions, each followed by a Rectified Linear Unit (ReLU) activation (*Arora et al., 2016*) and a max-pooling of 2 × 2 with a two stride for down-sampling. The number of feature channels is doubled in each step of down-sampling. The concatenation between the contracting and expanding paths uses two 3 × 3 convolution. Each block in an expansive path consists of 2 × 2 up-convolution, which reduces the number of channels by two, and two 3 × 3 convolution layers, each followed by a ReLU. The network's final layer is 1 × 1 convolution to map the feature map's size from 64 to the desired number of classes. The U-Net architecture utilizes the skip connections to compensates for lost information by passing information from encoder feature maps to the decoder portions at the same level.

In 2016, a group of researchers (*He et al., 2016*) developed the Residual Network (ResNet). ResNet is an efficient framework for CNNs. It was discovered that it could find

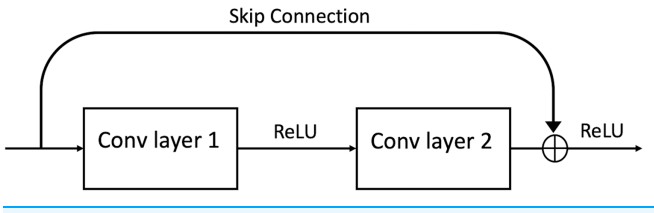

**Figure 3  A ResBlock within a ResNet.**     

an innovative solution to the problem of vanishing gradient through the "identity shortcut connection," which allows one or more layers to be skipped by taking activation from one layer and feeding it to another layer (see Fig. 3). ResNet has different variations in layers; 34-layer, 50-layer, 101-layer, 152-layer, and even 1,202-layer.

The Dense Convolutional Network (DenseNet) (*Huang et al., 2017*) is a logical extension of ResNet utilizing the shortcut/skip connections. The DenseBlock is different than ResBlock by using tensor concatenation so that with each skip connection, the network becomes denser. Therefore, in DenseNet, all DenseBlocks are connected directly to each other in a feed-forward style. Each block's input consists of all previous layers' feature maps by concatenating them. This architecture provides parameter and computational efficiency, vital information, and gradient flow that make networks easy to train and lead to implicit deep supervision.

*Xie et al. (2017)* developed a variation of ResNet, called ResNeXt, that follows the paradigm of split-transform-merge. ResNeXt introduces a hyper-parameter cardinality, which determines the number of independent paths to adjust the capacity of the model. The experiments show that increasing cardinality is more effective in gaining accuracy than increasing capacity (going deeper or wider).

In 2018, a research work (*Hu, Shen & Sun, 2018*) proposed the Squeeze-and-Excitation (SE) block to improve the network's representational power by enabling it to perform dynamic channel-wise features re-calibration. Notably, the SE block enables the network to learn the global weighting over all pixels in a channel and allows for fast information transferring between pixels in different image locations.

It is worth mentioning that there are three scaling dimensions of a CNN: depth, width, and resolution. Depth refers to how deep the network is (number of layers), width is how wide the network is (for example, the number of channels in a layer), whereas resolution refers to the image resolution. Theoretically, increasing the number of layers, channels, and image resolution should improve the network performance, but practically it doesn't follow. Therefore, in 2019, a group of researchers (*Tan & Le, 2019*) developed a novel architecture called EfficientNet. It is a lightweight convolutional neural network architecture based on AutoML (*Tan et al., 2019*) and a compound scaling method to systematically scale up depth, width, and image resolution. In the beginning, they developed a baseline network, EfficientNet-B0, based on a Neural Architecture Search (NAS) using the AutoML MNAS framework (*Tan et al., 2019*). After that, they scaled up the baseline to obtain a family of models from EfficientNet-B1 to EfficientNet-B7. EfficientNets can achieve better accuracy and computational efficiency over existing CNN

networks. The main building block is the mobile inverted bottleneck (MBConv) (*Tan et al., 2019*; *Sandler et al., 2018*). Also, they added the SE block for further improvements.

## RELATED WORKS

Image segmentation is an essential technique for analyzing and processing images by dividing the image into several significant parts depending on its pixel characteristics. Moreover, it is mostly considered the first and essential step in the medical image domain for many clinical applications as it provides detailed information on human body structures. The most general and early method of segmenting medical images is manual segmentation. Manual segmentation (*Fasihi & Mikhael, 2016*) is done by experts to extract the region of interest (ROI) with accurate boundary information. Even though this method is highly accurate, it has many weaknesses, such as taking a long time to get the result and needs trained experts. Furthermore, the segmentation's quality depends on the experts' experience and performance, which may lead in most cases to the difference in the segmentation results from expert to expert (*Heckel et al., 2013*). Prior deep learning era, there were and still, several methods and techniques that are developed and used for medical image segmentation, namely thresholding, region growing, deformable models, clustering, and classifiers. An overview of the different segmentation methods is shown in Table 1.

Thresholding (*Senthilkumaran & Vaithegi, 2016*) is a fast and straightforward segmentation method that segments an image by dividing pixels into several categories based on their intensity value. This method performs a sequential scan of the image pixels. If the image intensity is more significant than a specific threshold value, it is considered a foreground or part of the object. Otherwise, pixels are considered as background. This method was applied to successfully Ultrasonic images (*Wu, Songde & Hanqing, 1998*) and was used to create binary images to aid in the segmentation process of MRI brain tumors (*Murthy & Sadashivappa, 2014*). However, the thresholding method is sensitive to noise due to its inability to capture the image's spatial characteristics.

Region growing (*Masood et al., 2015*) is another segmentation technique to extract a connected region from an image based on a pre-defined criterion, such as intensity or other image features. It begins with selecting initial seeds, and then the regions grow by adding seed neighbors that satisfy the condition. This process repeats until no more pixels can be added to the region. The disadvantages of this method are its sensitivity to noise and seed initialization. Region growing has been used successfully for CT images (*Pohle & Toennies, 2001*) and is widely used for brain analysis, such as segmentation of brain tumors (*Deng et al., 2010*). Another segmentation technique is deformable models (*McInerney & Terzopoulos, 1996*), which are curves or surfaces that evolve to outline a goal object in the image using prior knowledge of object shape and under the influence of internal and external forces. Researchers in *Gupta et al. (1993)* proposed a system using deformable models to segment ventricle boundaries in cardiac MR images. This method provides robustness for noise, gaps, and other irregularities in objects' boundaries, although it requires manual interaction to tuning parameters.

**Table 1 Overview of medical image segmentation methods.**

| Reference | Method | Modality | Task | Advantages | Limitations |
|---|---|---|---|---|---|
| Thresholding | | | | | |
| (Wu, Songde & Hanqing, 1998) | Entropic | Ultrasonic | Breast and liver | Simple and fast | Sensitive to noise, and unable to segment most medical images |
| (Murthy & Sadashivappa, 2014) | Thresholding and morphological | MRI | Brain Tumor | | |
| Region growing | | | | | |
| (Pohle & Toennies, 2001) | Region growing | CT | Organs | Simple, less sensitive to noise, and separates regions fairly accurately | Sensitive to noise and seed values, and requires manual interaction |
| (Deng et al., 2010) | Adaptive region growing | MRI | Brain Tumor | | |
| Deformable | | | | | |
| (Gupta et al., 1993) | Deformable model | MRI | Cardiac | Robustness for noise | Requires manual tuning of parameters |
| Clustering | | | | | |
| (Yang et al., 2002) | FCM | MRI | Ophthalmology | Fast computation, easy for implementation, and work well for MRI | Does not utilize spatial information, and inability to work on CT images |
| (Wang et al., 2008) | FCM | MRI | Brain | | |
| Classifiers | | | | | |
| (Khalid, Ibrahim & Haniff, 2011) | KNN | MRI | Brain abnormalities | Robustness, easy to train, and work well for MRI and CT images | Rely on hand-crafted features, and time consuming |
| (Kalinin et al., 2005) | Decision Tree | CT | Anatomical regions | | |
| CNN | | | | | |
| (Ciresan et al., 2012) | Sliding-window CNN | Microscopy | Neuronal structures | Automatic feature extraction | Time-consuming process, and requires an amount of computing resources |
| (Havaei et al., 2017) | Cascaded CNN | MRI | Brain tumor | | |
| FCN | | | | | |
| (Ben-Cohen et al., 2016) | FCN | CT | Liver | Process varying input sizes and End-to-end segmentation | Loss of global context information and resolution |
| (Tran, 2016) | FCN | MRI | Cardiac | | |
| U-Net | | | | | |
| (Ronneberger, Fischer & Brox, 2015) | U-Net | Microscopy | Cell | Effectively capture localization and context information and effective with limited dataset images | Takes a long time to train because of a large number of parameters to learn |
| (Dong et al., 2017) | U-Net | MRI | Brain tumor | | |
| (Li et al., 2018) | H-DenseU-Net | CT | Liver lesion | | |
| (Kumar et al., 2018) | U-SegNet | MRI | Brain tissue | | |
| (Pravitasari et al., 2020) | UNet-VGG16 | MRI | Brain tumor | | |
| (Aboelenein et al., 2020) | HTTU-Net | MRI | Brain tumor | | |
| (Li et al., 2020) | MRBSU-Net | EUS | GIST | | |
| (Lou et al., 2020) | U-Net | CT | Esophagus | | |

Knowing that clustering methods (*Masood et al., 2015*) are unsupervised learning methods, these methods are based on statistical analysis to understand the distribution and representation of data. There are three main ways to segment medical images of the clustering approach: K-means, Fuzzy C-means (FCM), and Expectation-Maximization (EM). FCM algorithms are widely applied to MRI segmentation of different body parts

(*Yang et al., 2002*), especially the brain (*Wang et al., 2008*). These algorithms are simple and easy to understand, but they are sensitive to noise because they do not consider the data's spatial information. On the other hand, the classifiers method (*Masood et al., 2015*) is a supervised learning approach, and it is based on pattern recognition from the data. This method relies on extracting features from images using classical computer vision techniques such as Speeded Up Robust Features (SURF) (*Bay, Tuytelaars & Van Gool, 2006*), Scale Invariant Feature Transform (SIFT) (*Lowe, 1999*), etc. Then, features are fed to classification algorithms like Support Vector Machines (SVM) and KNearest Neighbours (KNN). Researchers in (*Khalid, Ibrahim & Haniff, 2011*) used kNN to segment brain abnormalities in MRI images, and in (*Kalinin et al., 2005*), they used a decision tree classifier to segmentation different anatomical regions in abdominal CT images. These methods are easy to train and simple. Still, they do not consider spatial information, and the data have to be segmented manually to be used in the automatic segmentation process.

The most recent successful medical image segmentation methods these days are based on deep learning techniques, especially CNN. An early approach to image segmentation using deep learning involved dividing the input image into several small segments. Then, feed each segment as input into deep neural networks and treat them as a classification problem to find the label of central pixels for each segment. This approach is mathematically expensive as it cannot process the entire image at once, so it should be repeated over the whole pixel array. Several previous works have applied deep learning in the medical image segmentation task. Researchers in *Ciresan et al. (2012)* used a sliding-window convolutional neural network to segment neuronal structures in stacks of electron microscopy (EM) images. Their approach has two drawbacks; it is time-consuming because each patch should be processed separately, which leads to redundancy of the computation and the inability of the network to learn global features. Moreover, researchers in *Havaei et al. (2017)* proposed a cascaded CNN architecture for brain tumor segmentation that is exploiting both local and global contextual features.

A group of researchers (*Long, Shelhamer & Darrell, 2015*) proposed a fully convolutional network (FCN), which is the first end-to-end semantic segmentation architecture that addresses the previously mentioned drawbacks. FCN architecture does not have any fully connected layers, allowing images of any size to be accepted. The output feature maps are up-sampled for producing dense pixel-wise output. It has been used in several medical applications and has attracted many researchers to explore this technique. In *Ben-Cohen et al. (2016)*, the researchers use FCN for liver segmentation and liver metastasis detection in CT examinations. Also, researchers in *Tran (2016)* used FCN for left and right ventricle segmentation in cardiac MRI. Furthermore, several state-of-the-art segmentation networks have been proposed based on FCN, such as PSPNet (*Zhao et al., 2017*), U-Net (*Ronneberger, Fischer & Brox, 2015*), DeepLab (*Chen et al., 2014*), and SegNet (*Badrinarayanan, Kendall & Cipolla, 2017*). Among these architectures, the U-Net has been proven to be one of the best frameworks for biomedical image segmentation as it won the Cell Tracking Challenge at ISBI 2015. For brain tumor segmentation in MRI images, the researchers in *Dong et al. (2017)* employed U-Net. In

contrast, in *Pravitasari et al. (2020)* they employed U-Net and VGG16 network in the encoder, and in *Aboelenein et al. (2020)* the researchers built a Hybrid Two-Track U-Net (HTTU-Net) by using Leaky Relu activation and batch normalization. In *Li et al. (2018)* they proposed a hybrid densely connected U-Net (H-DenseU-Net) that consisted of a densely connected network (DenseNet) and U-Net for automatic liver lesion segmentation from CT scans by replacing the convolutional layer with dense blocks in the encoder. A group of researchers (*Kumar et al., 2018*) proposed U-SegNet that is a hybrid of SegNet and U-Net segmentation architectures to improve automated brain tissue segmentation. A research work (*Li et al., 2020*) proposed multi-task refined boundary-supervision U-Net (MRBSU-Net) for gastrointestinal stromal tumor (GIST) segmentation from endoscopic ultrasound (EUS) images. Also, a group of researchers (*Lou et al., 2020*) employed a U-Net architecture with semiautomatic labeling to segment the esophagus from CT images.

Lung diseases, such as pneumonia, lung cancer, pneumothorax, are among the most common medical conditions globally. The COVID-19, which was first detected in late 2019, is also a lung disease caused by a novel coronavirus. The current research focuses on the medical condition of a pneumothorax. Still, there is a relation between pneumothorax and other lung diseases, such as COVID-19. Authors in *Martinelli et al. (2020)* described a pneumothorax as a complication and a notable clinical feature of Covid-19. Caution is required of the clinicians while evaluating patients, as they both share common symptoms of shortness of breath, which may force clinicians to search for alternative diagnoses (*Porcel, 2020*). The chest x-ray is a common and essential diagnostic method for both diseases. To detect COVID-19, several approaches have been proposed based on deep learning techniques. In *Elzeki et al. (2021)*, the authors used three pre-trained CNN models based on three different chest X-ray sets. Also, in *Kamal et al. (2021)*, the authors used eight pre-trained CNN models to detect COVID-19 from chest X-rays. Whereas in *Sheykhivand et al. (2022)*, the authors proposed a DNN model consisting of pre-trained CNN, Generative Adversarial Networks (GANS), and Long Short-Term Memory network (LSTM).

In 2019, a machine learning challenge was released on the Kaggle about detecting and localizing pneumothorax in radiographs, called SIIM-ACR Pneumothorax Segmentation. The number of participants in the challenge reached 1,457 teams in the development stage and 346 teams in the final stage. As shown in Table 2, the most successful methods for the Pneumothorax Challenge are based on deep CNNs.[1]

[dsmlkz] sneddy team, who took first place in the SIIM-ACR Pneumothorax Segmentation competition, used the U-Net architecture with pre-trained encoders. The encoders used were ResNet-34, ResNet-50, and SE-ResNext-50. All experiments except ResNet-50 were trained on 512 × 512 then up-trained on size 1,024 × 1,024 with a freezing encoder in early epochs. During the training, they used aggressive augmentation and a sample rate of images containing pneumothorax. As for post-processing, they applied a horizontal flip TTA and triplet threshold scheme.

X5 team, who took second place in the same challenge, divided their approach to classification and segmentation stages. In the classification stage, they trained the models

[1] The top 10 winning teams models of the Pneumothorax Challenge are available on https://siim.org/page/pneumothorax_challenge

**Table 2 Solutions and results for the top winning teams in the pneumothorax challenge.**

| Rank | Team | Network | Encoder | Techniques | Score |
|---|---|---|---|---|---|
| 1 | [dsmlkz] sneddy | U-Net | ResNet (34, 50), SE-ResNext-50 | Triplet threshold | 0.8679 |
| 2 | X5 | Deeplabv3+, U-Net | SE-ResNext (50, 101), EfficientNet (B3, B5) | Segmentation with Classification | 0.8665 |
| 3 | Bestfitting | U-Net | ResNet-34, SE-ResNext-50 | Lung segmentation and CBAM attention | 0.8651 |
| 4 | [ods.ai] amirassov | U-Net | ResNet-34 | Deep supervision | 0.8644 |
| 5 | Earhian | U-Net | SE-ResNext (50, 101) | ASPP and Semi-supervision | 0.8643 |

on all data to classify whether the image contained pneumothorax. While in the segmentation stage, they trained models only on data containing pneumothorax. The models used for classification were the U-Net model with SE-ResNeXt-50, SE-ResNeXt-101, and EfficientNet-b3 backbones, and for segmentation they used an averaged ensemble of U-Net model with SEResNeXt-50, SE-ResNeXt-101, EfficientNet-b3, and EfficientNetb5 backbones and DeepLabv3 model with ResNeXt-50 backbone. The size of the images used varies in the resolution of 768 × 768 or 1,024 × 1,024.

The bestfitting team took third place, used the U-Net model with ResNet-34 and SE-ResNext-50 backbones. First, U-Net with ResNet-34 has trained to segment the lungs in the Montgomery County X-ray Set (*Jaeger et al., 2014*). Then, they employed another U-Net with ResNet-34 as a pseudo-label model on the competition dataset to predict the CheXpert dataset (*Irvin et al., 2019*) and the National Institutes of Health (NIH) Chest X-Ray dataset (*Wang et al., 2017a*). Finally, they used the first U-Net model to crop the dataset images from 1,024 × 1,024 to 576 × 576 and 704 × 704 and fed them to U-Net with the SE-ResNext-50 model for segmenting pneumothorax. Also, they applied the attention mechanism by Convolutional Block Attention Module (CBAM) (*Woo et al., 2018*) on all models.

The fourth team is amirassov; they used the U-Net model with a deep supervision (*Wang et al., 2015*) branch to classify the empty mask and ResNet-34 as an encoder. For pre-processing, they reduced the image resolutions from 1,024 × 1,024 to 768 × 768, then applied a random crop of 512 × 512, and used data augmentation to increase the training set.

Earhian team ranked the fifth and used the U-Net model with Atrous Spatial Pyramid Pooling (ASPP) (*Chen et al., 2017*). The backbones SE-ResNeXt-50 and SE-ResNeXt-101 were trained on 1,024 × 1,024 resolution of images. The solution was based on the Mean Teacher Semi-Supervision method (*Tarvainen & Valpola, 2017*) with the NIH Chest X-ray dataset.

Additionally, Ayat/2ST-UNet team (*Abedalla et al., 2020*) proposed a Two-Stage Training system using U-Net (2ST-UNet) with a pre-trained ResNet-34 backbone. Their approach relied on training the network on lower resolution first, then retraining the network on high-resolution images. They also used data augmentation, TTA, and SWA techniques to improve the performance of their model. They found that Two-Stage training provided better convergence of the network.

The current work is proposing a new semantic segmentation model using the same dataset of the challenge. This model is based on U-Net architecture using different state-of-the-art deep CNN as encoders and the nearest-neighbor up-sampling in the decoder layer. Also, explore different techniques for training and prediction. Our method is highly accurate and differs from others by using less image resolution and less prediction time. It also applies an exhaustive search for the best and optimal combination of thresholds used and the weight averaging technique for the ensemble process of the models, where the weights are set automatically based on several experiments.

## METHOD

In this section, we describe our proposed framework for medical image segmentation. We build four semantic segmentation networks of the U-Net architecture with different backbones (B-UNets). Our design ensembles four B-UNets models (Ens4B-UNet) using weighted averaging to boost the segmentation performance. Furthermore, this section includes a detailed description of the dataset, a detailed explanation for our pre-processing steps and the used data augmentation techniques, a full discussion on the segmentation networks with robust encoders, and an illustration of post-processing techniques for generating accurate binary masks. An overview of the proposed framework pipeline for pneumothorax segmentation is shown in Fig. 4.

### Dataset description

This paper has used the chest X-ray dataset provided by the 2019 SIIM-ACR Pneumothorax Segmentation Challenge. The dataset consists of 12,047 training images with 12,954 masks and 3,205 test images. The round-2 test dataset's true masks are not provided, which means we have to submit our segmentation prediction results online on the Kaggle system that hosted the competition to evaluate our proposed model. The goal is to predict the pneumothorax's existence in the chest radiographic images, then predict the location and the extent of the pneumothorax condition in the images that have pneumothorax by creating accurate binary masks.

The images were provided in Digital Imaging and Communications in Medicine (DICOM) format (https://www.dicomstandard.org/), which is the most common standard for storing digital medical images and associated data. DICOM files consist of image data (pixels), metadata of image attributes, equipment configuration, and patient information, including name, ID, age, birth date, etc. Diagnosing pneumothorax is mainly based on X-rays, so we have focused on pixel image data. The size of each image in the dataset is 1,024 × 1,024 and has a single-channel grayscale.

The mask value for the images without pneumothorax is −1, while for the images with pneumothorax is a relative form of Run-Length Encoding (RLE). RLE is an efficient lossless compression format for storing binary masks by encoding foreground objects' location in segmentation. The RLE was provided as Encoded Pixels, a list of start pixels and the number of pixels after each of those starts included in the mask. The dataset was provided with a relative form of RLE, which means that the remaining pixel values are considered offsets after the first-pixel position. For example, 247981 1 1022 2 1021 3

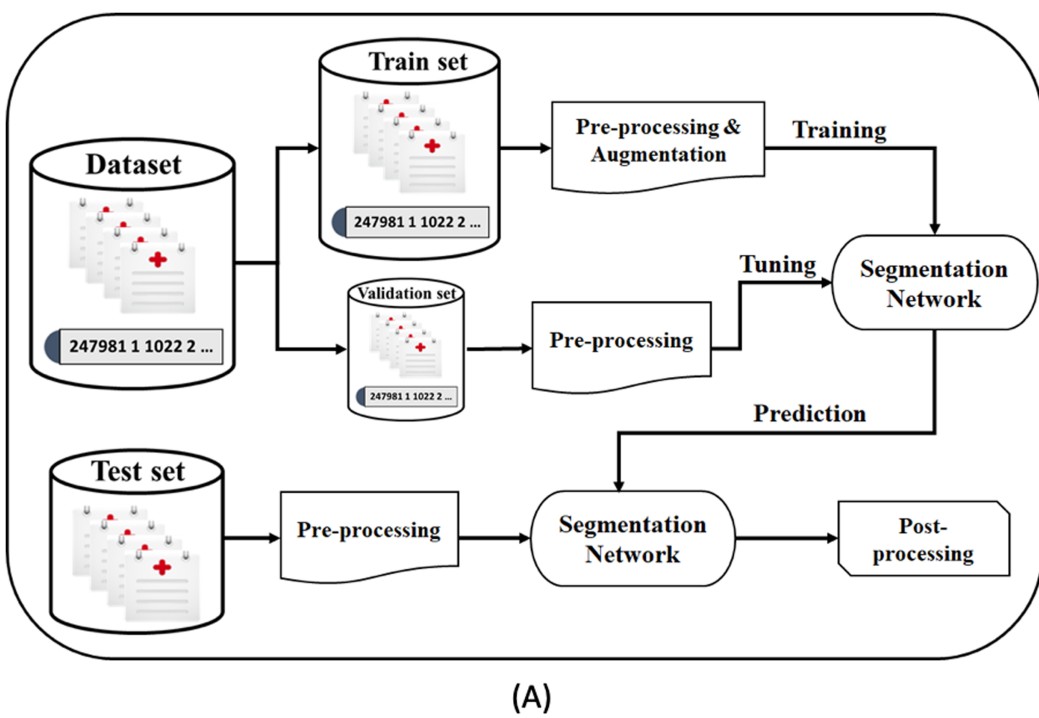

(A)

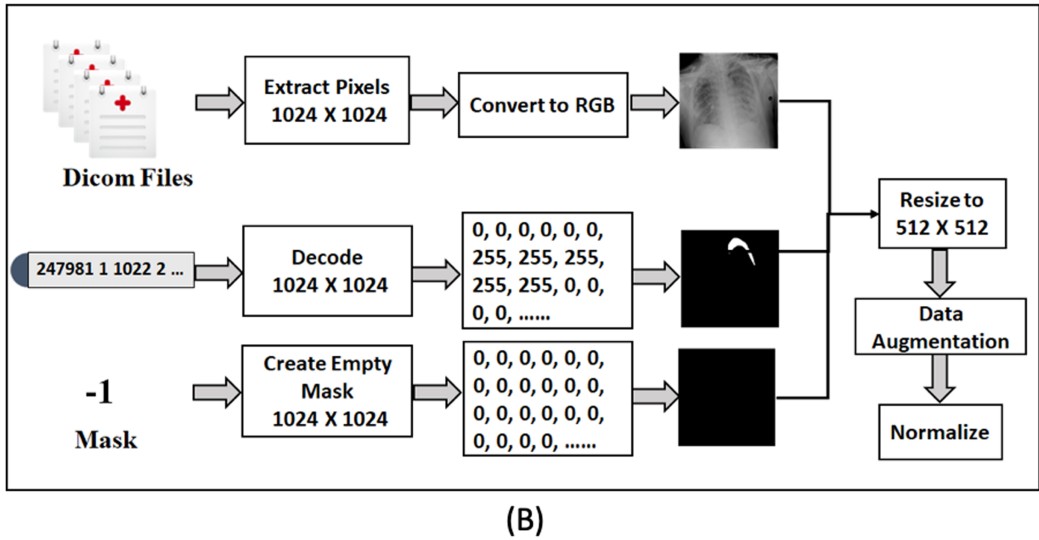

(B)

Figure 4 **The proposed methodology of pneumothorax segmentation.** (A) Overview of the proposed pipeline of pneumothorax segmentation. (B) Overview of the pre-processing steps of the DICOM files and Mask values.

means: take 1 pixel starting from 247,981, take 2 pixels starting from 249,004 after jumping 1,022 pixels from the previous run, and take 3 pixels after jumping 1,021 pixels of the prior run. Thus, the mask's pixels are as follows: 247,981, 249,004, 249,005, 250,027, 250,028, 250,029.

**Table 3 Dataset overview.**

| Attribute | Training set | Validation set |
|---|---|---|
| Number of cases | 10,842 | 1,205 |
| Number of positive cases | 2,405 | 264 |
| Number of negative cases | 8,437 | 941 |
| Number of cases has single masks | 1,853 | 192 |
| Number of cases has multiple masks | 552 | 72 |

Table 3 shows an overview of distribution in the training set and validation set. The dataset is unbalanced as it contains 2,669 cases of pneumothorax (22.15%), of which 2,045 cases have a single RLE mask, and 624 cases have multiple RLE masks. We randomly split the dataset into a 90% training set and a 10% validation set. The training set has 2,405 cases of pneumothorax (22.18%), of which 1,853 cases have a single RLE mask, and 552 cases have multiple RLE masks. The validation set also contains 264 cases of pneumothorax (21.91%), of which 192 cases have a single RLE mask, and 72 cases have multiple RLE masks.

## Data pre-processing and augmentation

Data pre-processing is an essential technique for improving data quality. First, we extract the pixel arrays from the DICOM files and then convert them into three red, green, and blue (RGB) channels to benefit from transfer learning. For the RLE mask, if the value of Encoded Pixels is −1, then we create an empty mask (black mask) with zero values of $1,024 \times 1,024$ size. Otherwise, we decode the Encoded Pixels to $1,024 \times 1,024$ black and white masks with 0 and 255 pixels values. Then, we have resized each image and its mask to $512 \times 512$ pixels. Figure 5 shows some examples of X-ray images, masks, and masks on the X-ray images to illustrate the dataset and how the affected areas look like. If the mask is entirely black, this indicates that it does not contain pneumothorax. In contrast, the white area in the mask indicates the area affected by the occurrence of pneumothorax.

The data augmentation technique has been used to make our proposed models more generalized and robust to overfitting. It can generate more data with different training patterns to help models in the learning process. For all models, we apply other augmentation techniques on the training dataset for image and mask together. That includes horizontal flip; rotations input up to 20 degrees by shift scale rotate, one of elastic transform, optical distortion, grid distortion, one of random contrast, random brightness, and random gamma. In addition to these transforms, we also have applied random crop parts of the image for the ResNet50-UNet model. Figure 6 shows an example of using augmentation operations.

Finally, we transform the images and their mask through normalization, which is an essential step in pre-processing medical images. We have normalized pixels intensity by rescaling all pixels in the images to a range from 0 to 1 instead of 0 to 255, and the pixels in the masks to 0 and 1 to get the binary pixel values (binary masks).

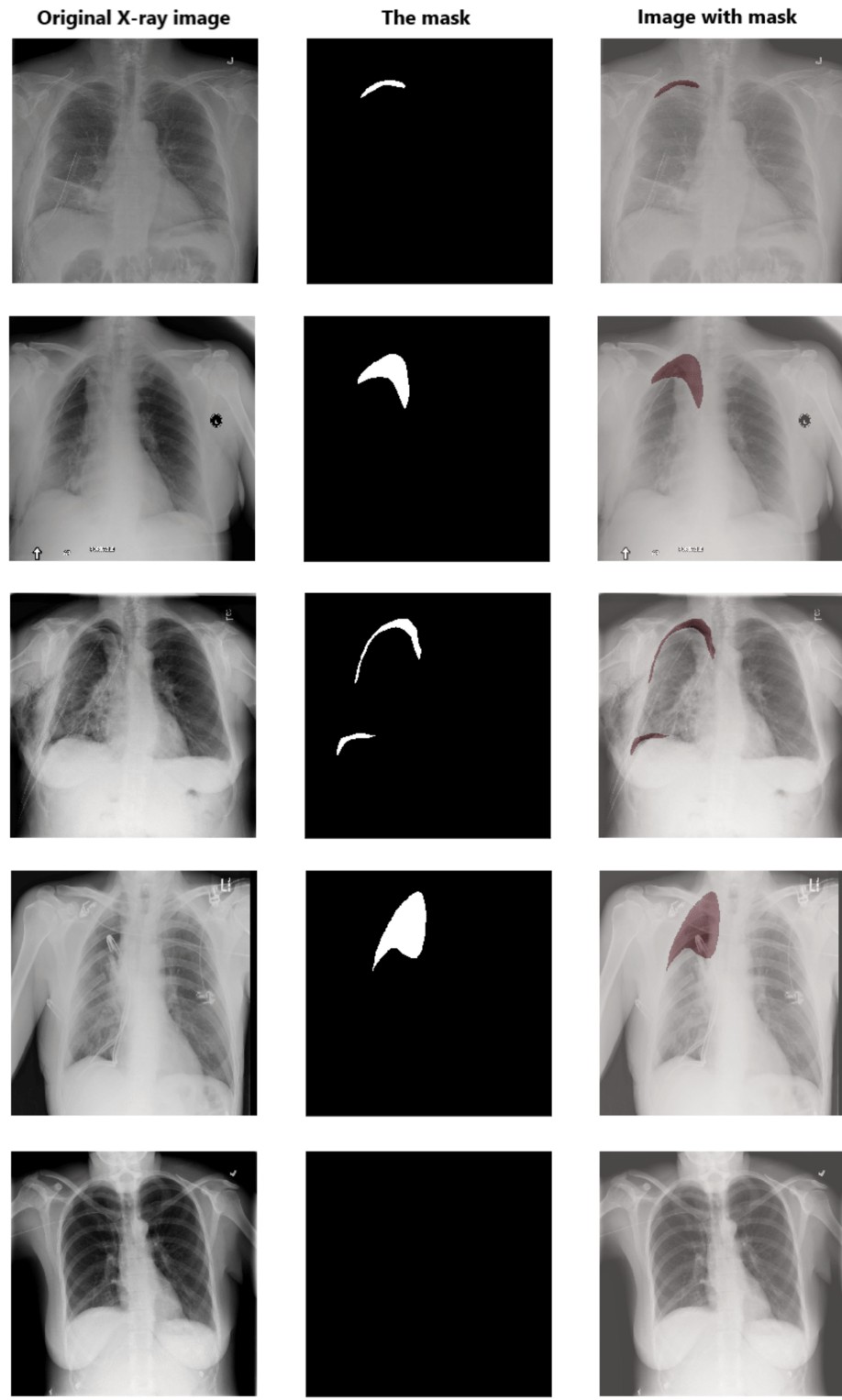

**Figure 5** Examples of some X-ray images (left), the masks (middle), and the X-ray images with masks (right).

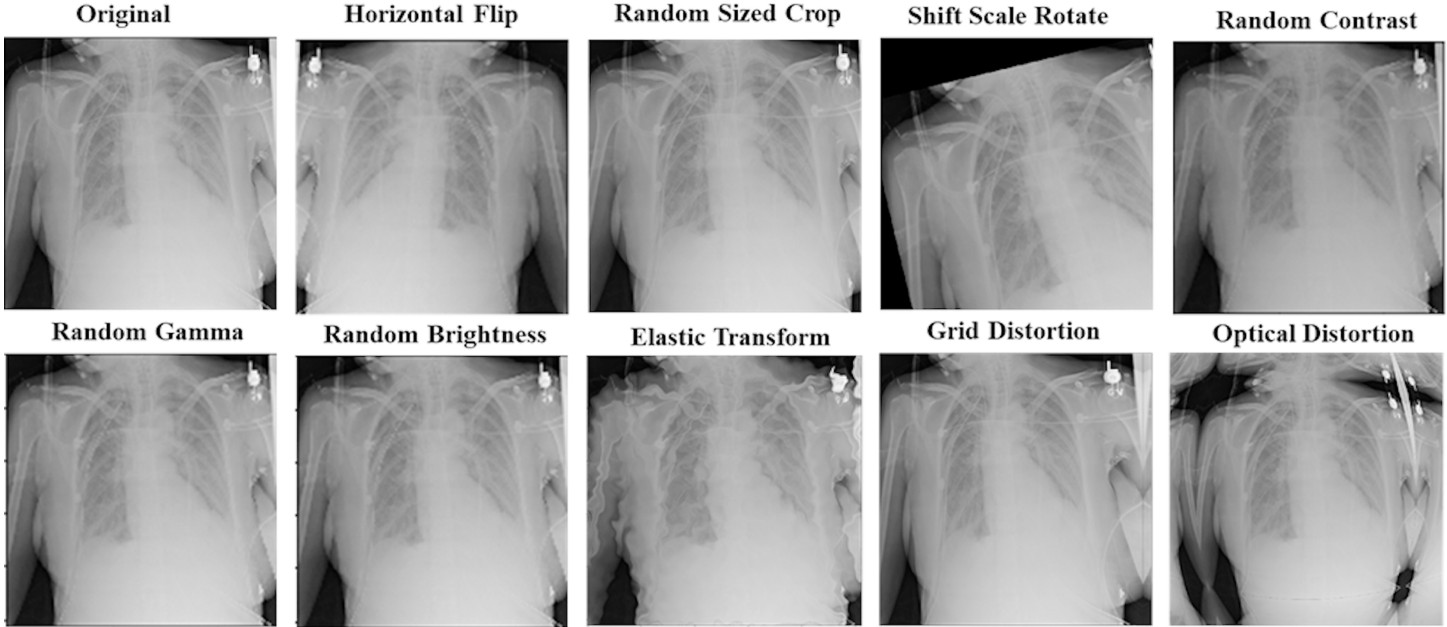

**Figure 6 An example of applying augmentation operations.**               

## Segmentation network architecture

U-Net is usually trained from scratch, starting with randomly initialized weights. Training from scratch is considered a difficult task as it is mathematically expensive and requires a large amount of data. On the other hand, transfer learning is a common computer vision technology used to initialize the network weights with better performance. The most common transfer learning way is to use pre-trained models. A pre-trained model is a model that was trained previously on a large benchmark dataset to provide a useful starting point to solve a new task that is different from the task that was created for. Therefore, in the design of our segmentation framework, we leverage the popular deep CNN models that pre-trained on the ImageNet dataset as backbones for our work instead of basic convolution blocks for feature extraction in the encoding path. The backbone refers to the base network architecture that takes the image as input and extracts the feature maps.

We propose four different B-UNet semantic segmentation architectures to segment the pneumothorax in chest X-ray images robustly. These four networks follow the well-known U-Net structure. Each network consists of an encoder (down-sampling path) to extract feature maps by capturing the image context and a decoder (up-sampling path) to expand the received feature maps for generating a pixel-wise segmentation mask. Moreover, each network includes skip-connections, which allow the transfer of information from the encoding path to the decoding path at the corresponding levels through the concatenation operation to obtain an accurate segmentation map. Figure 7 represents the architecture of our proposed semantic segmentation networks.

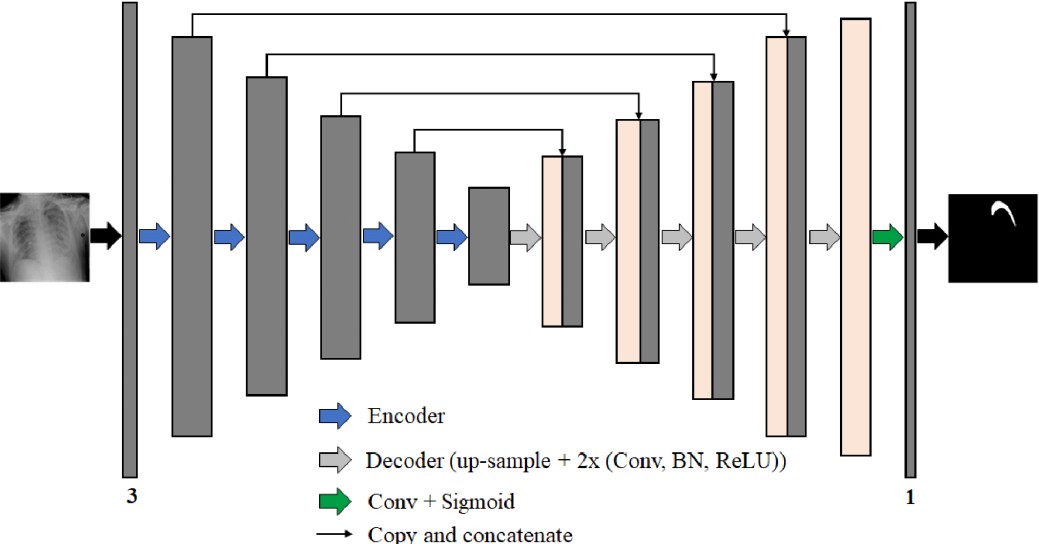

3                  1

➡️ Encoder

➡️ Decoder (up-sample + 2x (Conv, BN, ReLU))

➡️ Conv + Sigmoid

⟶ Copy and concatenate

**Figure 7 The architecture of the proposed semantic segmentation networks for pneumothorax segmentation from chest X-ray images.**

The four semantic segmentation networks are:

- ResNet50-UNet: the encoding path consists of 50-layer ResNet as a backbone network architecture.
- DenseNet169-UNet: the encoding path consists of 169-layer DenseNet as a backbone network architecture.
- SE-ResNext50-UNet: the encoding path consists of 50-layer ResNext with the SE module as a backbone network architecture.
- EfficientNetB4-UNet: the encoding path consists of EfficientNet-B4 as a backbone network architecture.

The backbone networks were originally designed for classification tasks. We modified them to accommodate the semantic segmentation task by removing the global average pooling layer and all the following fully connected layers from the end of each backbone network architecture. The decoder parts consist of five blocks in each of them, wherein each level of the first four blocks, we use the nearest-neighbor up-sampling layer to increase the image size by copying the value from the nearest pixel. The nearest-neighbor up-sampling output is then concatenated with the output of the corresponding part of the encoder. After that, we feed the feature map in two 3 × 3 convolutions, batch normalization (BN) (*Ioffe & Szegedy, 2015*), and ReLU activation layers. In the last decoder block, we use nearest-neighbor up-sampling followed by two 3 × 3 convolutions, BN, and ReLU activation layers. Except for DensNet169-UNet where the BN layer is not used in all decoder blocks. During decoding, the channel numbers are halved from 256 to 16. Finally, the output is obtained by applying a 3 × 3 convolution followed by a sigmoid activation (*Han & Moraga, 1995*). The segmentation network output is a pixel-by-pixel mask that defines the class of each pixel.

For ResNet50-UNet, we train the network for 60 epochs and a batch size of 10 using the stochastic gradient descent (SGD) optimizer with a momentum value of 0.9. For the learning rate, we apply cosine annealing scheduler (*Loshchilov & Hutter, 2016*) starting from 1e−3 and gradually decreasing to 1e−5. Also, we have used the SWA technique for the last five epochs to improve network generalization. For DenseNet169-UNet, we train the network for 80 epochs and a batch size of 6 using Adam optimizer. For the learning rate, we apply a cosine annealing scheduler starting from 1e−4 and gradually decreasing to 1e−6. The SWA technique is also used to average the last four epochs. For SE-ResNext50-UNet, we train the network for 80 epochs and a batch size of 6 using Adam optimizer. For the learning rate, we apply a cosine annealing scheduler starting from 1e−4 and gradually decreasing to 1e−6. The SWA technique is used to average the last three epochs. And finally, for EfficientNetB4-UNet, we train the network for 80 epochs and a batch size of 4 using Adam optimizer. For the learning rate, we apply a cosine annealing scheduler starting from 1e−4 and gradually decreasing to 1e−6. The SWA technique is used to average the last four epochs.

The loss function for training our models is the sum of the Binary Cross-Entropy (BCE) and Dice. BCE loss is defined as:

$$BCE = -1/N \sum_{i=1}^{N} \tag{1}$$

where $N$ is the number of samples, $y$ is the ground truth value, and $p$ is the predicted value. The Dice loss value is based on the Dice Similarity Coefficient (DSC) (*Dice, 1945*). The DSC works efficiently on the class imbalanced problems and is typically applied as a metric in image segmentation tasks to calculate the similarity between predicted pixels and the corresponding ground truth.

The formula of the Dice loss is defined as follows:

$$DSCL = 1 - \frac{2 \times |X \bigcap Y|}{|X| + |Y|} \tag{2}$$

where $X$ is the predicted set of pixel values, and $Y$ is the ground truth set of pixel values.

The formula for the loss function is obtained as follows:

$$BCE\text{-}Dice\ Loss = BCE + DSCL \tag{3}$$

## Post-processing and prediction ensembling

The Ens4B-UNet is a weighted average ensembling model for the outputs of the four proposed semantic segmentation networks. Ens4B-UNet refers to Ensembles 4 U-Net with Backbone networks. Each of the four segmentation networks in our framework contains a robust backbone and is shoa decoder inspired by the U-Net architecture. An overview of the proposed Ens4B-UNet framework wn in Fig. 8.

For all segmentation models, we apply TTA, where the original image in the test set is augmented with a horizontal flip transform. Then, the average prediction is calculated on the augmented images. The segmentation network output is an image with the same
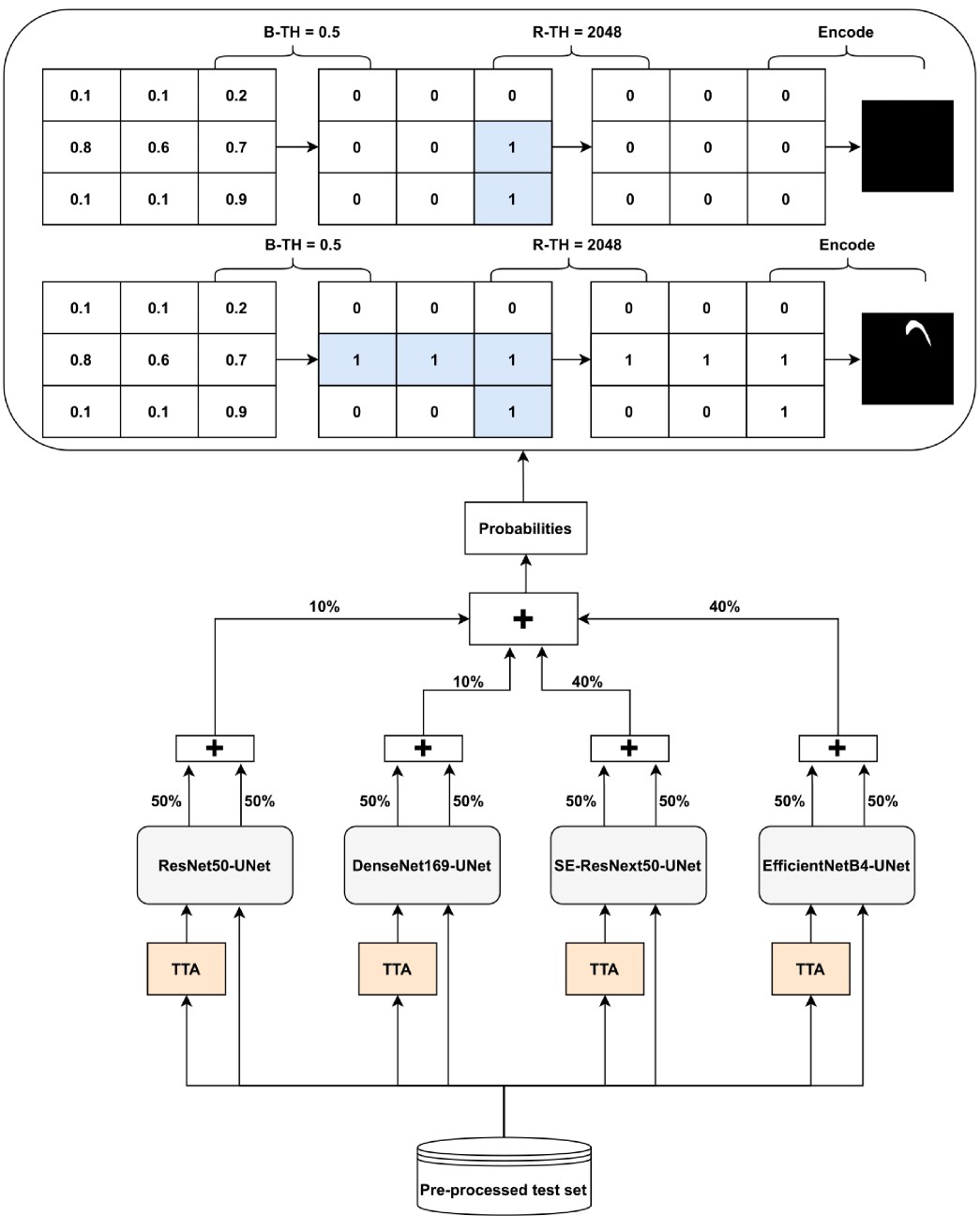

**Figure 8 Architecture of the Ens4B-UNet framework.**

size as each pixel's input image corresponds to a probability that belongs to a given class. To obtain the final binary mask, we start applying the binarization threshold (B-TH) to transform output probabilities to a discrete mask of zeros and ones. We set all pixel values above the B-TH to be one, while all other pixel values to be zero as depicted in Eq. (4).

$$M(x_i, y_i) = \begin{cases} 1 & \text{if } P(x_i, y_i) > B - TH \\ 0 & \text{if } P(x_i, y_i) \leq B - TH \end{cases} \tag{4}$$

**Table 4  Training parameters for all experiments.**

| Experiment | B-UNet Network | Optimizer | LR Schedule | Batch Size | Epochs |
|---|---|---|---|---|---|
| EXP1 | ResNet50-UNetSGD | 1e−3 to 1e−5 | 10 | 60 | |
| EXP2 | DenseNet169-UNet | Adam | 1e−4 to 1e−6 | 6 | 80 |
| EXP3 | SE-ResNext50-UNet | Adam | 1e−3 to 1e−5 | 4 | 100 |
| EXP4 | SE-ResNext50-UNet | Adam | 1e−4 to 1e−6 | 6 | 80 |
| EXP5 | SE-ResNext101-UNet | Adam | 1e−4 to 1e−6 | 4 | 100 |
| EXP6 | EfficientNetB3-UNet | Adam | 1e−4 to 1e−6 | 4 | 45 |
| EXP7 | EfficientNetB4-UNet | Adam | 1e−4 to 1e−6 | 4 | 80 |

where $M(x_i, y_i)$ is the segmented binary mask value associated with the pixel $(i, j)$ at the coordinates $(x_i, y_i)$, B-TH is the binarization threshold, and $P(x_i, y_i)$ is the pixel probability value.

After applying B-TH, we perform the removal threshold (R-TH) to remove small objects, which only accepts the objects whose pixel size is larger than R-TH. B-TH and R-TH's threshold values have been determined based on a grid search for different values using the validation dataset. The number of valuers for B-TH is 70 in the range from 0.2 to 0.9, with a step size of 0.01. The R-TH is defined as not using it or using a specific value, which is chosen by searching on a range of 1,024, 2,048, 3,072, and 4,096 pixels. Then, we multiply each pixel by 255 to get a predicted black and white mask. For the test set, we also resize the mask to 1,024 × 1,024 and encode it to RLE for submission.

To improve segmentation performance, we generate prediction maps by weight averaging ensemble of the four networks' outputs. Also, post-processing techniques have been used for the ensemble prediction map. We choose the weights based on the results of both the validation set and the test set. We find that the best performance achieves with the following weights: 40% of EfficientNetB4-UNet, 40% of SE-ResNext50-UNet, 20% of DenseNet169-UNet, and 20% of ResNet50-UNet.

# EXPERIMENTS

## Experiments setup
The proposed models have been implemented using well-known libraries in Python programming language, such as Keras (https://keras.io/) and Tensorflow (https://www.tensorflow.org/) as a backend. Also, we have used several Python packages like OpenCV (https://opencv.org/) for image processing, albumentations (*Buslaev et al., 2020*) the fast and flexible implementation for image augmentation, and pydicom (*Mason, 2011*) to work with DICOM files. All experiments have been performed on a workstation with NVIDIA Titan Xp GPU (12 GBs of memory), 32 GBs RAM, and Ubuntu 16.04 environment.

We have conducted several experiments with different configurations. Table 4 shows the parameters for training the experiments. For the first experiment (EXP1), we adopt ResNet-50 architecture as the U-Net segmentation network's backbone. For the second experiment (EXP2), we adopt DenseNet-169 architecture as the backbone of the U-Net

segmentation network. The SE-ResNext architecture has been adopted as the backbone of the U-Net segmentation network for the third experiment (EXP3) and the fourth experiment (EXP4) with 50-layer, and with 101-layer for the fifth experiment (EXP5). Finally, we adopt EfficientNet-B3 and EfficientNet-B4 architectures as the backbone of the U-Net segmentation networks for the sixth experiment (EXP6) and the seventh experiment (EXP7), respectively. We optimize the network parameters using the SGD for the first experiment (EXP1) and Adam for all other experiments. For the learning rate, we set a schedule from 1e−3 to 1e−5 for EXP1, EXP3, and EXP6, and from 1e−4 to 1e−6 for all other experiments. We have set the maximum batch size that can fit in memory space for each experiment. Also, We have applied the SWA technique for all experiments. For EXP4 and EXP6, it takes the average of the last three epochs. In EXP2, EXP3, EXP5, and EXP7, it takes the average of the last four epochs. For EXP1, it takes the average of the last five epochs.

## Evaluation metrics

To evaluate our segmentation networks, we first use the Intersection over Union (IoU) (*Jaccard, 1912*) metric during training and tuning of both the training and validation sets. IoU is the area of overlap between the ground truth ($P_{\text{true}}$) and the predicted segmentation ($P_{\text{predicted}}$) divided by the area of union between them. The following formula computes the IoU:

$$IoU(P_{\text{true}}, P_{\text{predicted}}) = \frac{P_{\text{true}} \bigcap P_{\text{predicted}}}{P_{\text{true}} \bigcup P_{\text{predicted}}} \qquad (5)$$

Also, we use the accuracy, recall, precision, and F-measure (F1-Score) to evaluate our model on a validation dataset.

$$\text{Accuracy} = \frac{TP + TN}{TP + FP + FN + TN} \qquad (6)$$

$$\text{Precision} = \frac{TP}{TP + FP} \qquad (7)$$

$$\text{Recall} = \frac{TP}{TP + FN} \qquad (8)$$

$$\text{F-measure} = \frac{2TP}{2TP + FP + FN} \qquad (9)$$

Then, we use the peak signal-to-noise ratio (PSNR) (*Fardo et al., 2016*) metric in decibels (dB) which defined via the mean squared error (MSE). The PSNR between ground truth mask $I$ and predicted mask $\hat{I}$ is defined as:

$$\text{PSNR} = 20 \cdot \log_{10} \left( \frac{MAX_I}{\sqrt{\frac{1}{N} \sum_{i=1}^{N} \left( I(i) - \hat{I}(i) \right)^2}} \right) \qquad (10)$$

**Table 5 The number of trainable parameters, training time per epoch, training step time, and IoU score of the validation set for all proposed B-UNet networks.**

| Experiment | B-UNet network | #Params | Train step time (ms) | Train time/epoch | IoU |
|---|---|---|---|---|---|
| EXP1 | ResNet50-UNet | 32.5 M | 568 | 10.3 min | 0.7535 |
| EXP2 | DenseNet169-UNet | 19.5 M | 437 | 13.2 min | 0.7669 |
| EXP3 | SE-ResNext50-UNet | 34.5 M | 472 | 21.3 min | 0.6986 |
| EXP4 | SE-ResNext50-UNet | 34.5 M | 644 | 19.4 min | 0.7820 |
| EXP5 | SE-ResNext101-UNet | 59.89 M | 709 | 32 min | 0.7795 |
| EXP6 | EfficientNetB3-UNet | 17.77 M | 431 | 19.5 min | 0.7589 |
| EXP7 | EfficientNetB4-UNet | 25.6 M | 516 | 23.3 min | 0.7758 |

where *MAX* is the maximum pixel value equal to 255 for 8-bit pixel encoding. Finally, we use the DSC provided by the pneumothorax competition in the testing and predicting phase on the test set. The following formula computes the DSC.

$$\text{DSC}\,(X, Y) = \frac{2 \times |X \bigcap Y|}{|X| + |Y|} \tag{11}$$

where $X$ is the predicted set of pixels and $Y$ is the ground truth. This metric is defined to be 1 when both $X$ and $Y$ are empty and the final score is the mean DSC for each image in the test set.

## RESULTS AND DISCUSSION

Table 5 summarizes the size of model components in terms of trainable parameters in millions, training step time, training time per epoch, and IoU score of the validation set for each proposed semantic segmentation network in our proposed model. For ResNet50-UNet, each training step takes about 568 mS, and each epoch takes around 10.3 min. It achieves an IoU score of 0.7535. For DenseNet169-UNet, an IoU score of 0.7669 is obtained, with each training step takes about 437 ms and each epoch takes nearly 13.2 min. For SE-ResNext50-UNet in EXP3, each training step takes about 472 ms, each epoch takes approximately 21.3 min, and an IoU score of 0.6986 is achieved. For SE-ResNext50-UNet in EXP4, each training step takes about 644 ms, each epoch takes about 19.4 min, and an IoU score of 0.7820 is achieved. For SE-ResNext101-UNet in EXP5, each training step takes about 709 ms, each epoch takes about 32 min, and an IoU score of 0.7795 is achieved. For EfficientNetB3-UNet, each training step takes about 431 ms, each epoch takes about 19.5 min, and an IoU score of 0.7589 is achieved. For EfficientNetB4-UNet, each training step takes about 516 ms, each epoch takes approximately 23.3 min, and an IoU score of 0.7758 is achieved.

Based on these results, we have chosen the following experiments to continue with the tuning and prediction phases. First, we have chosen ResNet50-UNet from EXP1 for its fastest convergence during training. Then, DenseNet169-UNet from EXP2 achieves a higher score with fewer parameters compared to ResNet50-UNet. Also, SE-ResNext50-UNet from EXP4 has been selected as it achieves a higher score than EXP3 and a higher score with fewer parameters and less time than SE-ResNext101-UNet from EXP5. It is also

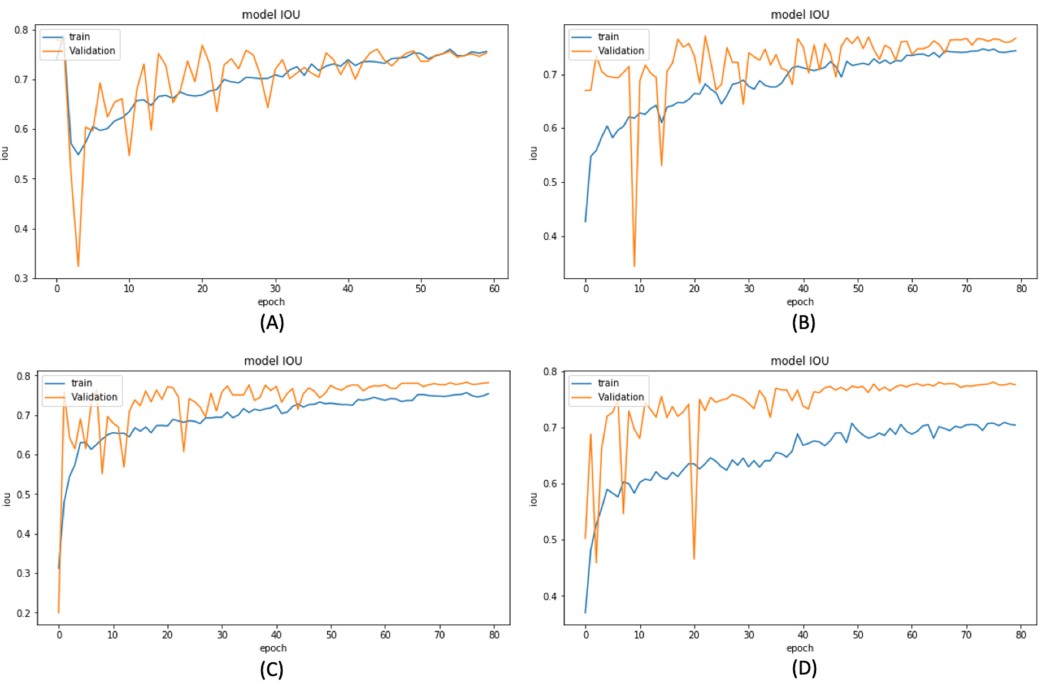

**Figure 9 Training and validation IoU evaluation over training of four proposed segmentation network.** (A) ResNet50-UNet (B) DenseNet169-UNet (C) SE-ResNext50-UNet (D)EfficientNetB4-UNet.

shown that the lower learning rate in EXP4 helps the model's learning process compared to EXP3. Finally, EfficientNetB4-UNet from EXP7 has been selected as it achieves a higher score than EfficientNetB3-UNet from EXP6. Figures 9 and 10 illustrate the IoU and loss plots of the training and validation sets over training epochs for our four segmentation networks.

Figure 11 demonstrates the confusion matrix of the proposed segmentation networks on the validation set. Table 6 shows the accuracy, recall, precision, and F-measure of our segmentation networks with 0.5 of B-TH on the validation set. We have evaluated each model of our segmentation framework on the validation set using different configurations. The configurations include using or not using horizontal flip TTA, using R-TH from automatic search or not using it, and an automatic search on B-TH value. Table 7 shows the performance of our segmentation networks on the validation set in terms of IoU with different configurations.

For ResNet50-UNet, the worst IoU result (0.7542) is gained using only B-TH; however, adding TTA improves the result to 0.7714. Whereas the use of R-TH with B-TH but without TTA achieves IoU of 0.7968, adding TTA improves the outcome to 0.7981. The same happened to the rest segmentation networks, as the DenseNet169-UNet has the worst IoU results using only B-TH (0.7656) while using R-TH with B-TH and adding TTA improves the result to 0.8007. Regarding SE-ResNext50-UNet, the use of only B-TH achieves IoU of 0.7832, but using R-TH with B-TH and adding TTA improves the result to 0.8000. Furthermore, the EfficientNetB4-UNet has achieved 0.7786 as IoU score using only

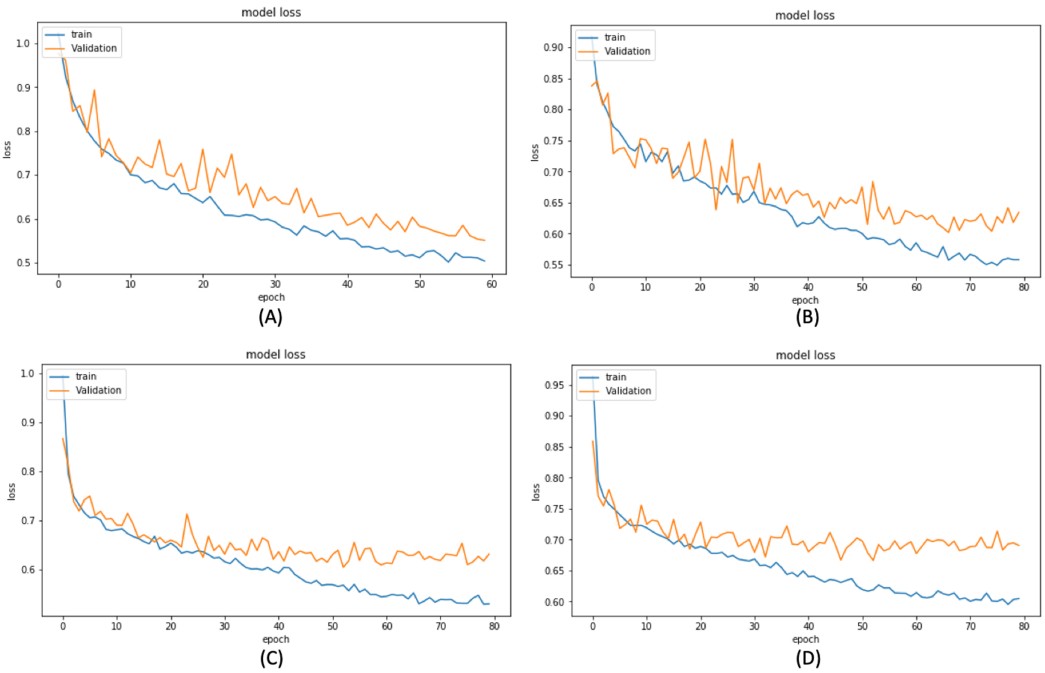

**Figure 10 Training and validation loss evaluation over training of four proposed segmentation network.** (A) ResNet50-UNet. (B) DenseNet169-UNet. (C) SE-ResNext50-UNet. (D) EfficientNetB4-UNet.

B-TH, but using R-TH with B-TH and adding TTA improves the result to 0.8030. We clearly can see that the use of TTA and R-TH have improved the predictions for all models.

It is worth mentioning that The EfficientNetB4-UNet has achieved the highest IoU score of 0.8030 with a horizontal flip TTA, automatic search for B-TH with a value of 0.55, and R-TH with a pixels value of 2,048. The DenseNet169-UNet obtains the next highest result with an IoU score of 0.8007, B-TH of 0.56, and R-TH of 1,024 pixels. Then the SE-ResNext50-UNet with IoU score of 0.8000, B-TH with a value of 0.56, and R-TH of 1,024 pixels. Finally, the ResNet50-UNet has the lowest IoU score of 0.7981, B-TH of 0.73, and R-TH of 2,048 pixels. Also, Table 8 shows PSNR results for validation masks using optimal values of B-TH.

For each segmentation model, we have selected the best configurations based on the validation set results to predict an unseen test set. Table 9 shows the performance of the different models in the proposed segmentation framework on the test set in terms of mean DSC. These models have used the optimal B-TH and R-TH values obtained from the validation set. It is inferred that EfficientNetB4-UNet outperforms other models with a mean DSC of 0.8547. Followed by SE-ResNext50-UNet with a score of 0.8515, then DenseNet169-UNet with a score of 0.8473, followed by the ResNet50-UNet, a score of 0.8400. Knowing that all models have powerful results, we have been encouraged to ensemble all proposed models' predictions by the weighted averaging technique. We have chosen the weights based on the results of both the validation set and the test set. We have experimented with the weights, and the best performance is achieved using the

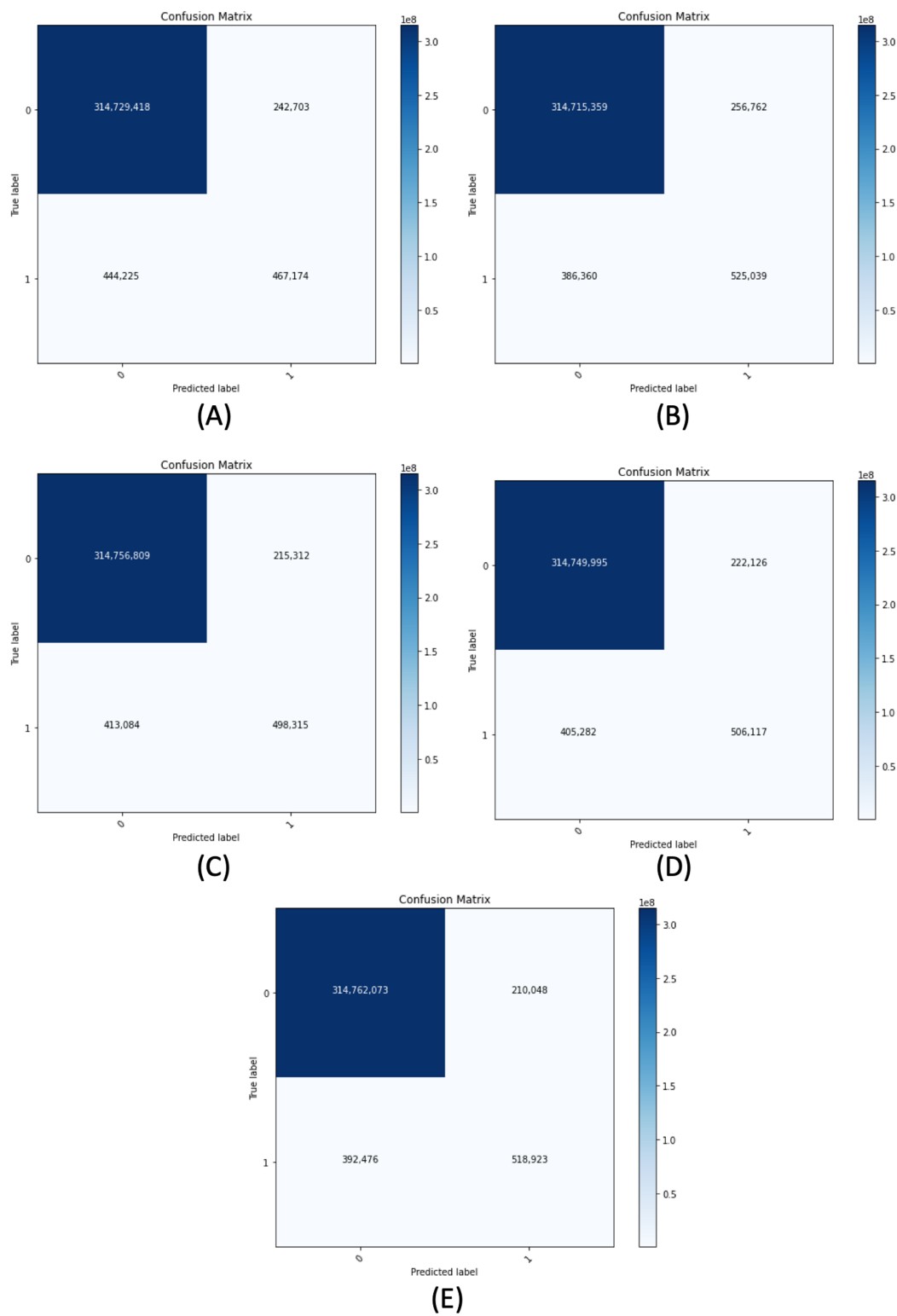

**Figure 11 Confusion matrix for all proposed segmentation network for the validation dataset.**
(A) ResNet50-UNet. (B) DenseNet169-UNet. (C) SE-ResNext50-UNet. (D) EfficientNetB4-UNet.
(E) Ens4B-UNet.

**Table 6 Pixel-wise classification results of our segmentation networks on the validation set.**

| Model | Accuracy (%) | Recall (%) | Precision (%) | F-measure (%) |
|-------|-------------|-----------|--------------|---------------|
| ResNet50-UNet | 99.78 | 51.26 | 65.81 | 57.63 |
| DenseNet169-UNet | 99.80 | 57.61 | 67.16 | 62.02 |
| SE-ResNext50-UNet | 99.80 | 54.68 | 69.83 | 61.33 |
| EfficientNetB4-UNet | 99.80 | 55.53 | 69.50 | 61.74 |
| Ens4B-UNet | 99.81 | 56.94 | 71.19 | 63.27 |

**Table 7 The IoU result of the validation set using different base networks.**

| Segmentation network | TTA | B-TH | R-TH | IoU |
|---------------------|-----|------|------|-----|
| ResNet50-UNet | ✗ | Auto 0.86 | ✗ | 0.7542 |
| | ✗ | Auto 0.87 | Auto 2048 | 0.7968 |
| | ✓ | Auto 0.77 | ✗ | 0.7714 |
| | ✓ | Auto 0.73 | Auto 2048 | 0.7981 |
| DenseNet169-UNet | ✗ | Auto 0.70 | ✗ | 0.7656 |
| | ✗ | Auto 0.83 | Auto 3072 | 0.7989 |
| | ✓ | Auto 0.89 | ✗ | 0.7759 |
| | ✓ | Auto 0.65 | Auto 2048 | 0.8007 |
| SE-ResNext50-UNet | ✗ | Auto 0.82 | ✗ | 0.7832 |
| | ✗ | Auto 0.81 | Auto 3072 | 0.7986 |
| | ✓ | Auto 0.89 | ✗ | 0.7920 |
| | ✓ | Auto 0.56 | Auto 1024 | 0.8000 |
| EfficientNetB4-UNet | ✗ | Auto 0.90 | ✗ | 0.7786 |
| | ✗ | Auto 0.21 | Auto 2048 | 0.8015 |
| | ✓ | Auto 0.59 | ✗ | 0.7867 |
| | ✓ | Auto 0.55 | Auto 2048 | 0.8030 |

**Table 8 The PSNR (dB) results of our segmentation networks on the validation set.**

| Model | PSNR (dB) |
|-------|-----------|
| ResNet50-UNet | 26.71 |
| DenseNet169-UNet | 27.01 |
| SE-ResNext50-UNet | 27.05 |
| EfficientNetB4-UNet | 27.08 |
| Ens4B-UNet | 27.20 |

following weights; 40% of EfficientNetB4-UNet, 40% of SE-ResNext50-UNet, 20% of DenseNet169-UNet, and 20% of ResNet50-UNet. Table 10 shows some of the experiments on the validation set for different weights. Additionally, the ensemble technique has achieved a mean DSC of 0.8608, which is considered from the top 1% of the pneumothorax solutions that can achieve this significant result.

**Table 9 Average prediction time per sample, and prediction mean DSC score reported for different proposed segmentation models evaluated on the test set.**

| Model | Prediction time (ms) | DSC score leaderboard | |
|---|---|---|---|
| | | Public | Private |
| ResNet50-UNet | 37.2 | 0.9113 | 0.8400 |
| DenseNet169-UNet | 56 | 0.9041 | 0.8473 |
| SE-ResNext50-UNet | 66.3 | 0.9014 | 0.8515 |
| EfficientNetB4-UNet | 53.6 | 0.9065 | 0.8547 |
| Ens4B-UNet | – | 0.9060 | 0.8608 |

**Table 10 Different weights for the proposed models on the validation set.**

| ResNet50-UNet (%) | DenseNet169-UNet (%) | SE-ResNext50-UNet (%) | EfficientNetB4-UNet (%) | IoU |
|---|---|---|---|---|
| 40 | 40 | 10 | 10 | 0.7796 |
| 30 | 30 | 20 | 20 | 0.7845 |
| 25 | 25 | 25 | 25 | 0.7867 |
| 20 | 20 | 30 | 30 | 0.7875 |
| 10 | 10 | 40 | 40 | 0.7877 |
| 10 | 10 | 30 | 50 | 0.7844 |
| 10 | 10 | 20 | 60 | 0.7858 |
| 10 | 20 | 30 | 40 | 0.7857 |

**Table 11 Comparisons with most recent current work on the pneumothorax medical condition.**

| Reference | Dataset size | Task | Results |
|---|---|---|---|
| (*Mostayed, Wee & Zhou, 2019*)* | 12,047 | Segmentation | Official DSC 76.04 |
| (*Jakhar, Bajaj & Gupta, 2019*) ** | 12,047 | Segmentation | DSC 84.3 |
| (*Jun, Kim & Kim, 2018*) | 100,000 | Classification | AUC 0.911 |
| (*Luo et al., 2019*) | 11,051 | Segmentation | MPA 0.93, DSC 0.92 |
| Ens4B-UNet* | 12,047 | Segmentation | Official DSC 0.8608 |

**Notes:**
* indicates that the reference works on the same dataset that we do.
** means that the reference work result is not on the official test set but the test set divided from the training set.

Table 11 shows a comparison between our approach performance and the other works on pneumothorax segmentation and classification. In *Mostayed, Wee & Zhou (2019)* they employed content-adaptive (*Su et al., 2019*) convolution instead of concatenation operations for the skip connections in the U-Net architecture and tested their approach to the pneumothorax segmentation task. Their approach achieved a 76.04% mean dice coefficient on the official test set of the SIIM-ACR Pneumothorax Segmentation Challenge. In *Jakhar, Bajaj & Gupta (2019)* they proposed a segmentation model pneumothorax segmentation. The model is based on U-Net with pre-trained ResNet as a backbone and

**Table 12 The comparison between our proposed model and different teams results on the Pneumothorax segmentation challenge.**

| Top (%) | Rank | Team/Model | Image size | DSC score leaderboard | |
|---|---|---|---|---|---|
| | | | | **Public** | **Private** |
| 1 | 1 | [dsmlkz] sneddy | 1,024 | 0.8985 | 0.8679 |
| | 5 | Earhian | 1,024 | 0.9035 | 0.8643 |
| | 13 | Ens4B-UNet | 512 | 0.9060 | 0.8608 |
| 2 | 22 | [ods.ai] 11,111 good team is all you need | 512, 1,024 | 0.9024 | 0.8557 |
| 3 | 31 | [ods.ai] Vasiliy Kotov | 1,024 | 0.8995 | 0.8537 |
| | 33 | imedhub ppc64 | 1,024 | 0.9093 | 0.8532 |
| 4 | 58 | OmerS | 1,024 | 0.9180 | 0.8477 |
| 6 | 80 | Mohamed Ramzy | 768 | 0.9124 | 0.8451 |
| 9 | 124 | Ayat/2ST-UNet (2) | 256, 512 | 0.9023 | 0.8356 |
| | 127 | DataKeen | 512 | 0.9073 | 0.8353 |
| 19 | 279 | diCELLa | – | 0.9041 | 0.8096 |
| 21 | 308 | Marsh | – | 0.6215 | 0.7757 |
| 24 | 340 | pete | – | 0.6351 | 0.6983 |

evaluated on the chest X-ray dataset from SIIM-ACR Pneumothorax Segmentation Challenge. In *Jun, Kim & Kim (2018)* they proposed an ensemble of 50-layer CNN model with different sizes of chest X-rays. They resized $1,024 \times 1,024$ images to $512 \times 512$, $384 \times 384$, and $256 \times 256$ for each of these networks. They evaluated their model on the Chest X-ray dataset (*Wang et al., 2017b*) containing more than 100,000 chest radiography and achieved a magnitude of 0.911 of Area Under the ROC Curve (AUC). In *Luo et al. (2019)*, the researchers proposed the fully convolutional DenseNet (FC-DenseNet) with a spatial and channel squeeze, excitation module (scSE), and a multi-scale module for pneumothorax segmentation. They evaluated their approach on 11,051 chest X-rays and achieved 0.93 of Mean Pixel-wise Accuracy (MPA) and 0.92 of DSC.

Table 12 shows a comparison between the performance of our approach and the other teams' methods in the pneumothorax challenge on the same test set. It indicates that our method achieves competitive results with a tiny difference of 0.0071 away from the team ranked first in the pneumothorax segmentation challenge. It is worth noting that the large image size has a significant impact on improving performance for this task. On the other hand, large images need more computation operations per layer and more memory requirements and training time. Our model has chosen to use less image resolution with less prediction time on choosing more image resolution and more prediction time.

## CONCLUSION AND FUTURE WORK

This paper has proposed a medical image segmentation framework called Ens4B-UNet, which is composed of four encoder-decoder networks based on the U-Net architecture with pre-trained backbone networks as encoders, and the nearest-neighbor up-sampling in

the decoders. The four networks are ResNet50-UNet, DenseNet169-UNet, SE-ResNext50-UNet, and EfficientNetB4-UNet. All the networks have used batch-normalization except for the DenseNet169-UNet decoder. Moreover, we have applied SWA for the training procedure and used an effective data augmentation technique to avoid the overfitting problem. We have applied horizontal flip TTA and different optimal thresholds, including B-TH and R-TH, for post-processing. We have evaluated our models' performance on the chest X-rays dataset from the 2019 SIIM-ACR Pneumothorax Segmentation Challenge. It is worth mentioning that Ens4B-UNet achieves a mean DSC of 0.8608, which is considered one of the top 1% results in the competition.

Due to computational resource limitations, all the Ens4B-UNet framework's segmentation networks have been trained on $512 \times 512$ images. In the future, we aim to train our networks on the full-sized resolution to utilize all information. Also, we would like to extend and improve our method by working on other medical segmentation datasets. It is also worthy of developing our study to apply our approach to Coronavirus Disease 2019 (COVID-19) X-ray images shortly.

### Funding
This work was supported by the Deanship of Research at the Jordan University of Science and Technology via Grant 20190180 and NVIDIA Corporation donated the Titan Xp GPU used for this research. The funders had no role in study design, data collection and analysis, decision to publish, or preparation of the manuscript.

### Grant Disclosures
The following grant information was disclosed by the authors:
Jordan University of Science and Technology: 20190180.
NVIDIA Corporation.

### Competing Interests
The authors declare that they have no competing interests.

### Author Contributions
- Ayat Abedalla conceived and designed the experiments, performed the experiments, performed the computation work, prepared figures and/or tables, and approved the final draft.
- Malak Abdullah conceived and designed the experiments, performed the experiments, analyzed the data, performed the computation work, prepared figures and/or tables, and approved the final draft.
- Mahmoud Al-Ayyoub conceived and designed the experiments, performed the experiments, analyzed the data, authored or reviewed drafts of the paper, and approved the final draft.
- Elhadj Benkhelifa conceived and designed the experiments, analyzed the data, authored or reviewed drafts of the paper, and approved the final draft.

## Data Availability

The data is available at a public Kaggle competition:

https://www.kaggle.com/c/siim-acr-pneumothorax-segmentation/overview.

The code is available at GitHub:

https://github.com/Ayat-Abedalla/Ens4B-UNet.

## Supplemental Information

Supplemental information for this article can be found online at http://dx.doi.org/10.7717/peerj-cs.607#supplemental-information.

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
