# Peer review of "Chest X-ray pneumothorax segmentation using U-Net with EfficientNet and ResNet architectures"

_PeerJ Computer Science, doi:10.7717/peerj-cs.607_

## Round 0.1 · original submission · Major Revisions

Based on the comments received from the reviewers, the paper can be considered for a revision should the authors address the concerns raised by the reviewers. The revised version shall address all the comments made by the reviewers and shall highlight the changes made with a point by point response rebuttal.

Reviewer 1 ·

Basic reporting

The written manuscript is unambiguous. However, English should be revised and corrected further. Most of the errors are grammatically related.
And Specifically:
Too many words repetitions:
- Repetition and redundancy can cause problems at the level of either the entire paper or individual sentences.
- Use a variety of different transition words.
- Vary the structure and length of your sentences.
- Don't use the same pronoun to reference more than one antecedent.
Some sentences are too long: - Break up your sentences
Sentence Repetition:
- Avoid repetition at the sentence level
- Avoid sentences that restate the main point of the previous sentence. - Don’t restate points you’ve already made.
The introductory sections show the context, nevertheless, there should be a clear proposition for the research objective at the abstract section. Structure is clear, figures and tables are well stated and represented. Raw data are provided with the required descriptive metadata identifiers.

Experimental design

The novelty of the research is obvious as it proposes a novel end-to-end semantic segmentation model, named Ens4B-UNet, for medical images as Ensembles 4 U- Net architectures with pre-trained Backbone networks. All underlying data have been provided; they are robust, statistically sound, & controlled. The overall performance and results are accepted .

Validity of the findings

The research falls within the scope of the journal. The objectives and methodologies are clear and well defined. The used algorithms and methods are described with sufficient detail & information to replicate.
Conclusions and results are well stated with the required diagrams.
However, the authors are required to provide the Confusion matrix of the validated images.

·

Basic reporting

Proofreading is required. There are a lot of places with punctuation errors, capital letters usage, unnecessary question marks, and redundant latex tags.
Diagrams could be made visually more appealing especially Figure 14 and Figure 10.

Experimental design

The paper does add to the body of knowledge. It aptly lists out the research gap, targets a specific issue, and gives sufficiently enough design and experimentation.

Validity of the findings

The authors are working on a dataset from the public domain and are also sharing their standing against the other works done to solve the same problem. This confirms the validity of their work. They aptly link the conclusions to the claims made.

Additional comments

no comment

·

Basic reporting

This paper is well organized and written. The paper proposes end-to-end semantic
segmentation model, named Ens4B-UNet, for medical images as Ensembles 4 U-Net
architectures with pre-trained model.

I recommend the change of the title to be reduced. I suggest "Chest x-ray pneumothorax segmentation using U-Net and Efficient ResNet architecture.

The introduction needs to be reduce. Have only the main parts and concentrate the motivation and contributions.

Experimental design

-The Experimental results need more details about the segmented images.
-Investigate the IoU score in details.
-Statistical calculations are required such as variance, mean, median, and the resulting histograms.
-SNR, PSNR, Error rates are required to be determined for the segmented images.

Validity of the findings

-The confusion matrix parameters like accuracy , F1-score, Recall, Should be calculated.
-The Hyper parameter values should be decelerated.
- The output of the Curve results from the program needs to be investigated and shown.

Additional comments

-The related work need to enhanced more recent works should be listed in higher top journals.
-The introduction should be summarized.
-Experimental results need to be realized and illustrate the confusion matrix parameters like accuracy , F1-score, Recall, Should be calculated.- The Hyper parameter values should be decelerated.
- The output of the Curve results from the program needs to be investigated and shown

---

## Round 0.2 · Minor Revisions

Based on the recommendations made by the reviewers in the revised reviews, I am glad to share that the paper can almost be considered for publication. Please address the minor comments from the reviewers before the paper can be published.

·

Basic reporting

minor proof reading is required

Experimental design

no comment

Validity of the findings

no comment

·

Basic reporting

well addressed

Experimental design

well addressed

Validity of the findings

Authors should be updated their introduction section by adding the most recent published papers and compare with the proposed methodology in the paper.
1- Kamal, K. C., Yin, Z., Wu, M., & Wu, Z. (2021). Evaluation of deep learning-based approaches for COVID-19 classification based on chest X-ray images. Signal, Image and Video Processing, 1-8.
2-Elzeki OM, Shams M, Sarhan S, Abd Elfattah M, Hassanien AE. 2021. COVID-19: a new deep learning computer-aided model for classification. PeerJ Comput. Sci. 7:e358 DOI 10.7717/peerj-cs.358
3-Sheykhivand, S., Mousavi, Z., Mojtahedi, S., Rezaii, T. Y., Farzamnia, A., Meshgini, S., & Saad, I. (2021). Developing an efficient deep neural network for automatic detection of COVID-19 using chest X-ray images. Alexandria Engineering Journal, 60(3), 2885-2903.

Additional comments

need minor revision

---

## Round 0.3 · accepted · Accept

Congratulations on the article acceptance. Please follow the journal policy on the publication process. It is the responsibility of the authors to ensure that the article has no contents from other published material which may lead to any potential copyright violation and that the article adhere to the publisher's policy on copyright material.